

# The influence of rhizosphere soil fungal diversity and complex community structure on wheat root rot disease

Xuejiang Zhang[1,2,3,*], Heyun Wang[4,*], Yawei Que[1,2,3], Dazhao Yu[1,2,3] and Hua Wang[1,2,3]

[1] Hubei Key Laboratory of Crop Disease, Insect Pests and Weeds Control, Wuhan, Hubei Province, China
[2] Key Laboratory of Integrated Pest Management on Crops in Central China, Ministry of Agriculture, Wuhan, Hubei Province, China
[3] Institute of Plant Protection and Soil & Fertilizer, Hubei Academy of Agricultural Sciences, Wuhan, Hubei Province, China
[4] HuBei University of Technology, Wuhan, Hubei Province, China
* These authors contributed equally to this work.

Corresponding authors
Dazhao Yu, dazhaoyu1956@126.com
Hua Wang, wanghua4@163.com

## ABSTRACT

Wheat root rot disease due to soil-borne fungal pathogens leads to tremendous yield losses worth billions of dollars worldwide every year. It is very important to study the relationship between rhizosphere soil fungal diversity and wheat roots to understand the occurrence and development of wheat root rot disease. A significant difference in fungal diversity was observed in the rhizosphere soil of healthy and diseased wheat roots in the heading stage, but the trend was the opposite in the filling stage. The abundance of most genera with high richness decreased significantly from the heading to the filling stage in the diseased groups; the richness of approximately one-third of all genera remained unchanged, and only a few low-richness genera, such as *Fusarium* and *Ceratobasidium*, had a very significant increase from the heading to the filling stage. In the healthy groups, the abundance of most genera increased significantly from the heading to filling stage; the abundance of some genera did not change markedly, or the abundance of very few genera increased significantly. Physical and chemical soil indicators showed that low soil pH and density, increases in ammonium nitrogen, nitrate nitrogen and total nitrogen contributed to the occurrence of wheat root rot disease. Our results revealed that in the early stages of disease, highly diverse rhizosphere soil fungi and a complex community structure can easily cause wheat root rot disease. The existence of pathogenic fungi is a necessary condition for wheat root rot disease, but the richness of pathogenic fungi is not necessarily important. The increases in ammonium nitrogen, nitrate nitrogen and total nitrogen contributed to the occurrence of wheat root rot disease. Low soil pH and soil density are beneficial to the occurrence of wheat root rot disease.

## INTRODUCTION

Soil microbial diversity is important to sustainable agriculture because microbes mediate many processes that are essential to the agricultural productivity of soil (*Lupwayi, Rice & Clayton, 1998*). However, to meet the food demand of an increasing population, intensive agricultural practices and excessive cultivation of crops have destroyed soil structure and ignored the biological potential of roots or rhizospheres to efficiently mobilize and acquire soil nutrients (*Parkinson & Coleman, 1981*; *Schreiner & Bethlenfalvay, 1996*; *Assaf, Turk & Ameed, 2009*; *Kumar & Pratush, 2014*; *Taheri, Hamel & Gan, 2015*; *Ai et al., 2015*; *Rashida et al., 2016*). As such, soils have very low biological activity, and plants growing in these soils are predisposed to soil-borne pathogens (*Sivasithamparam, 1993*; *Kirkegard et al., 2008*; *Wintera, Mol & Tiedemann, 2014*; *Almasudy, You & Barbetti, 2015*).

With the destruction of soil structure, the degradation of soil and the increase in soil-borne pathogens, wheat, as one of the three major staple foods in the world, grown in Asia (China), Australia, Europe, North America, and South America, is vulnerable to attack by a complex of root pathogens, which results in tremendous yield losses (*Duffy, 2000*). Annual losses in wheat industries due to soil-borne fungal pathogens amount to over billions of dollars worldwide (*Paulitz, Smiley & Cook, 2002*; *Mavrodi et al., 2012*; *Okubara, Dickman & Blechl, 2014*). All cultivars of wheat are attacked by several soil-borne fungal pathogens that cause root diseases (*Mavrodi et al., 2012*). The primary fungal pathogens include the following: *Fusarium culmorum*, *F. pseudograminearum*, *Gaeumannomyces graminis* var. *Tritici*, *Bipolaris sorokiniana*, and *Alternaria* spp. in *Ascomycota*; *Rhizopus oryzae*, *Rhizoctonia solani* and *Penicillium* spp. in *Basidiomycota*; *Pythium* spp. in *Oomycota*; and *Curvularia* spp. in *Deuteromycota* (*Mielke, 1998*; *Wintera, Mol & Tiedemann, 2014*). Root diseases are difficult to control because these soil-borne fungi are ubiquitous, the pathogens often occur as a complex (*Paulitz & Adams, 2003*; *Mavrodi et al., 2012*), and they can easily survive on infected plant debris or form durable chlamydospores in the soil with or in the absence of growing hosts and can outgrow or evade plant defenses (*Smith et al., 2003*; *Wintera, Mol & Tiedemann, 2014*). There are no resistant varieties among adapted cultivars of wheat and no chemical controls, although certain seed treatments can provide some early benefits to seedling health (*Paulitz & Scott, 2006*; *Davis et al., 2008*).

Therefore, it is very important to study the relationship between soil microbial diversity and plants to understand the occurrence and development of crop root rot. Plants depend on the ability of roots to communicate with rhizosphere soil microorganisms through signaling pathways, creating a connection between plants and microorganisms (*Meena et al., 2013*; *Li et al., 2016*; *Kumar et al., 2017*). The composition of the rhizosphere soil microbiota can negatively or positively influence plant traits such as stress tolerance, health, development, and productivity (*Kristin & Miranda, 2013*; *Lakshmanan, Selvaraj & Bais, 2014*; *Miao et al., 2016*). The plant, in turn, cultivates the structural and functional diversity of microbial communities in the rhizosphere soil by adjusting soil pH, releasing secondary metabolites into the rhizosphere soil (*Chakraborty et al., 2011*;

*Meena, Rakshit & Meena, 2016*; *Kumar et al., 2017*), reducing competition for beneficial microbes, and providing an energy source, mostly in the carbon-rich rhizosphere soil (*Davis et al., 2008*; *Rashida et al., 2016*). Nutrients are also drivers for rhizosphere soil community structure. Soil-plant-microbial health must remain in equilibrium to maintain sustainable agricultural practices (*Kumar et al., 2017*; *Narula, Anand & Dudeja, 2013*; *Ramırez-Bahena et al., 2013*). Under unfavorable conditions, some fungi can cause plant diseases and sometimes even total loss of crop yields (*Miao et al., 2016*).

In addition to agrochemicals (*Handiseni et al., 2013*), fertilization (*Phillips & Fahey, 2007*; *Ai et al., 2015*), soil types (*Buyer, Roberts & Russek-Cohen, 1999*; *Rasche et al., 2006*), tillage (*Lupwayi, Rice & Clayton, 1998*) and crop rotation (*Kirkegard et al., 2008*; *Wintera, Mol & Tiedemann, 2014*), which can influence rhizosphere soil microorganisms, the structure and function of rhizosphere soil microbiota may also be affected by the plant physiology of different plant genotypes (*Söderberg, Olsson & Bååth, 2002*; *Rasche et al., 2006*) and may also fluctuate among the vegetation stages of the same plant genotype (*Gyamfi et al., 2002*; *Rasche et al., 2006*). Plant growth stage influences root physiology and changes the quality and quantity of root exudates; consequently, these changes select for different root-associated microorganisms at different growth stages (*Dunfield & Germida, 2003*; *Houlden et al., 2008*; *Li et al., 2014*). The purpose of this study was to study the diversity of fungi, the variation in community structure and the trends in the microbial species in the rhizosphere soil of healthy and diseased wheat roots at different heading and filling stages of wheat growth. Additionally, by combining these results with the physical and chemical properties of soil, the possible causes of wheat root rot disease were revealed, providing an important theoretical basis and practice for the improved control of wheat root rot disease.

## MATERIALS & METHODS

### Rhizosphere soil sampling

Field experiments were approved by the Research Council of the Institute of Plant Protection and Soil & Fertilizer, Hubei Academy of Agricultural Sciences (project number:17.035.18).

The Institute of Plant Protection and Soil & Fertilizer, Hubei Academy of Agricultural Sciences granted Ethical approval to carry out the study within its facilities (Ethical Application Ref: hb375-a6c3d).

In Xiangyang Original Farm, from wheat filling stage in early April to dough stage in May, the physiological growth of wheat plants changed rapidly, and the root exudates changed sharply from April to May, which were important factors influencing the great changes of rhizosphere microorganisms. In addition, the rapid change of temperature is also an important factor. The temperature is around 17 degrees in early April and 24 degrees in early May.

From 2010 to 2016, we conducted the investigation and efficacy control test of wheat root rot disease in Xiangyang Original Farm for six consecutive years. According to our investigation and experiment, we found that the root rot of wheat plantation in

A

B

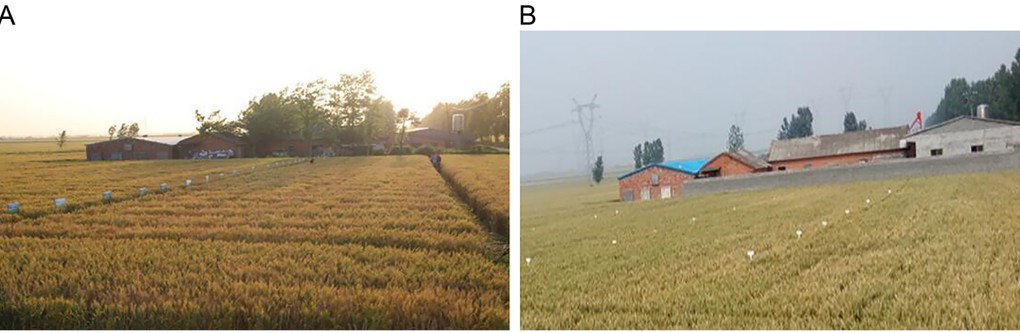

**Figure 1 Sampling plots of wheat root.** This picture mainly shows the readers the serious situation of the root rot disease in the wheat field of Xiangyang Original Farm. After the occurrence of wheat root rot disease, the ears of wheat is becoming abnormal white, and even false ripening in advance. Judging by this phenotype, when the roots of wheat plants were dug out, it was observed that the diseased roots were unusually brown, dark brown, and even black compared to the healthy roots. These diseased roots can lead to plant death at a later stage, leading to crop failure. (A) The occurrence of wheat root rot in this wheat field in 2015 and 2016. This land has a total of three acres, and 17 plots have been set up to test the efficacy of wheat root rot. The disease index of the wheat root rot reached 43%. (B) The soil sample collection of wheat root in this wheat field in 2017. Areas with diseased wheat and healthy wheat were marked by inserting white cards at fixed points. The samples, in April and May, were collected from the areas labeled in April.

Xiangyang Original Farm occurred seriously (the disease index reaches 43% Fig. 1A), and the main pathogenic factors were continuous cropping obstacle, large temperature variation, rice and wheat crop rotation. The main root rot occurred in the wheat field of Xiangyang Original Farm was caused mainly by *Gaeumannomyce gramim* (Sacc.) Arx et Olivier, *Pellicularia Rolfsii* (Sacc.) West, *Rhizoctonia cerealis* and *F. oxysporum* F.S.P. Niveum.

The symptoms of wheat root rot in April were not particularly obvious, so multiple rhizosphere soil samples were collected from multiple points. Then, we defined and selected rhizosphere soil samples collected in April by the sample area with the most significant symptoms of wheat root rot in May (Fig. 1B). The sampling method consisted of first investigating the occurrence of wheat root rot, selecting weak seedlings and sampling the brown parts of the root as the diseased wheat rhizosphere. Areas with diseased wheat and healthy wheat (the diameter of each area was not more than 10 m) were marked by inserting cards at fixed points at least 10 m apart. Five samples were collected from the healthy and diseased rhizosphere soil respectively in May, then we selected five samples from the corresponding healthy and diseased wheat area respectively in April.

The roots of the whole plant and the soil on the roots were collected. The majority of soil on the roots was shaken off, and samples were collected for the determination of physical and chemical soil properties. The rhizosphere soil samples wheat plants were placed in a sealed pocket and quickly stored in a dry-ice box for dry-ice preservation. After all the samples were collected, they were immediately brought to the laboratory and stored in a cryogenic refrigerator for future use.

## Extraction and PCR amplification of total genomic DNA from rhizosphere soil fungi

The soil attached to the root was brushed off, a 0.1-g soil sample was accurately weighed, and the total genomic DNA from all samples was extracted according to the instructions provided with an E.Z.N.A.® soil DNA Kit (Omega Biotek, Norcross, GA, USA). Total DNA was detected by 1% agarose gel electrophoresis, and the purity and concentration of DNA were determined with NanoDrop 2000 UV-vis spectrophotometer (Thermo Scientific, Waltham, MA, USA). All DNA samples were stored in a refrigerator at −20 °C. Fungal diversity was determined by amplifying the ITS1 region using the ITS1F and ITS2R primer sets for fungi. The primer sequences were ITS1F 5′-CTTGGTCATT TAGAGGAAGTAA-3′ and ITS2R 5′-GCTGCGTTCTTCATCGATGC-3′, and the amplification conditions were predenaturation at 95 °C for 5 min, 27 cycles of 95 °C for 30 s, 55 °C for 30 s, and 72 °C for 45 s, and elongation at 72 °C for 5 min. Three replicates of the PCR were performed, and a 20-μL reaction system (4 μL of 5× FastPfu buffer solution, 2 μL of 2.5 mM dNTPs, 0.8 μL of primer (5 μM), 0.4 μL of FastPfu polymerase and 10 ng of fungal total genomic DNA) was used.

## Illumina HiSeq2500 sequencing

Purified amplicons were pooled in equimolar and paired-end sequenced on an Illumina MiSeq PE300 platform/NovaSeq PE250 platform (Illumina, San Diego, CA, USA) according to the standard protocols by Majorbio Bio-Pharm Technology Co. Ltd. (Shanghai, China). The raw reads were deposited into the NCBI Sequence Read Archive (SRA) database (accession number: PRJNA549031).

## Data processing

The sequencing depth was more than 30,000 original reads per library. Raw fastq files were demultiplexed, quality-filtered by Trimmomatic (*Bolger, Lohse & Usadel, 2014*) and merged by FLASH 1.2.7 (*Magoč & Salzberg, 2011*) with the following criteria: (i) The 300 bp reads were truncated at any site receiving an average quality score of <20 over a 50 bp sliding window, and the truncated reads shorter than 50 bp were discard, reads containing ambiguous characters were also discarded; (ii) only overlapping sequences longer than 10 bp were assembled according to their overlapped sequence. The maximum mismatch ratio of overlap region is 0.2. Reads that could not be assembled were discarded; (iii) Samples were distinguished according to the barcode and primers, and the sequence direction was adjusted, exact barcode matching, two nucleotide mismatch in primer matching.

## OTU and species community analysis

Operational taxonomic units (OTUs) with 97% similarity cutoff were clustered using UPARSE version 7.1 (*Edgar, 2013*; *Stackebrandt & Goebel, 1994*) and chimeric sequences were identified and removed using UCHIME. The taxonomy of each OTU representative sequence was analyzed by RDP Classifier version 2.2 (*Wang et al., 2007*) against the Unite (Release 7.0 http://unite.ut.ee/index.php) database using confidence threshold of

**Table 1  The analysis of alpha diversity index in healthy and diseased groups.**

| Sample/Estimators | Sobs | Ace | Chao | Shannon | Coverage |
|---|---|---|---|---|---|
| H4 | 129.40 | 142.21 | 142.14 | 3.03 | 0.9994 |
| D4 | 135.60 | 154.71 | 149.52 | 3.37 | 0.9995 |
| H5 | 163.40 | 178.98 | 180.19 | 3.41 | 0.9993 |
| D5 | 155.80 | 160.04 | 161.46 | 3.32 | 0.9997 |

70%. Finally, the effective tag data, low-frequency tag data, annotated tag data, and OTU data obtained from each sample were counted by a script. Additionally, we used R software to calculate the annotation ratio of OTUs and each taxonomic level and the relative abundance of the species in each taxonomic level.

Based on the above valid OTU data, the following evolutionary analysis was carried out: (a) evolutionary relationships and relative abundance information of species systems based on OTU data in samples were determined, and species annotation results for a single sample were visualized using KRONA software (http://sourceforge.net/projects/krona); (b) related genera were selected, a phylogenetic tree for the OTUs of these genera was constructed (QIIME software package: make_phylogeny.py: http://qiime.org/scripts/make_phylogeny.html), and the systematic evolutionary relationship was displayed by combining the relative abundance of OTUs and the reliability of annotation using a Perl script; and (c) local Perl scripts were used to select the dedicated OTUs for intrasample and intersample phylogenetic analysis and to compare relative abundance.

To analyze the community structure of species, relative abundance thermograms were plotted at the OTU level and the genera level by R software. Alpha diversity was analyzed using mothur software. The data of soil physical and chemical properties was analyzed by SPSS 16.0 software. Cluster analysis and principal coordinate analysis (PCoA) were also carried out to compare samples.

## RESULTS

### Fungal diversity in the rhizosphere soil of wheat root

According to the diversity index, the community richness and diversity of the diseased groups were higher than those of the healthy groups in the heading stage, and there was a significant difference in community diversity between the diseased group and the healthy group (Shannon index: $P < 0.01$) (Table 1). High community richness can also be a factor in disease suppression. However, the community richness and diversity of the disease group and the healthy group in the filling stage were higher than those of the disease group and the healthy group in the heading stage, and there was a significant difference between the disease group and the healthy group in the heading stage (Sobs index: $P < 0.05$; Shannon index: $P < 0.05$). The community richness and diversity of the healthy group in the filling stage were higher than those of the diseased group in the filling stage.

### Community structure of fungi in the rhizosphere soil

There were 1,393 OTUs distributed among six phyla, 346 genera and 549 species. The six phyla were *Ascomycota*, *Basidiomycota*, *Zygomycota*, *Chytridiomycota*, *Glomeromycota*,

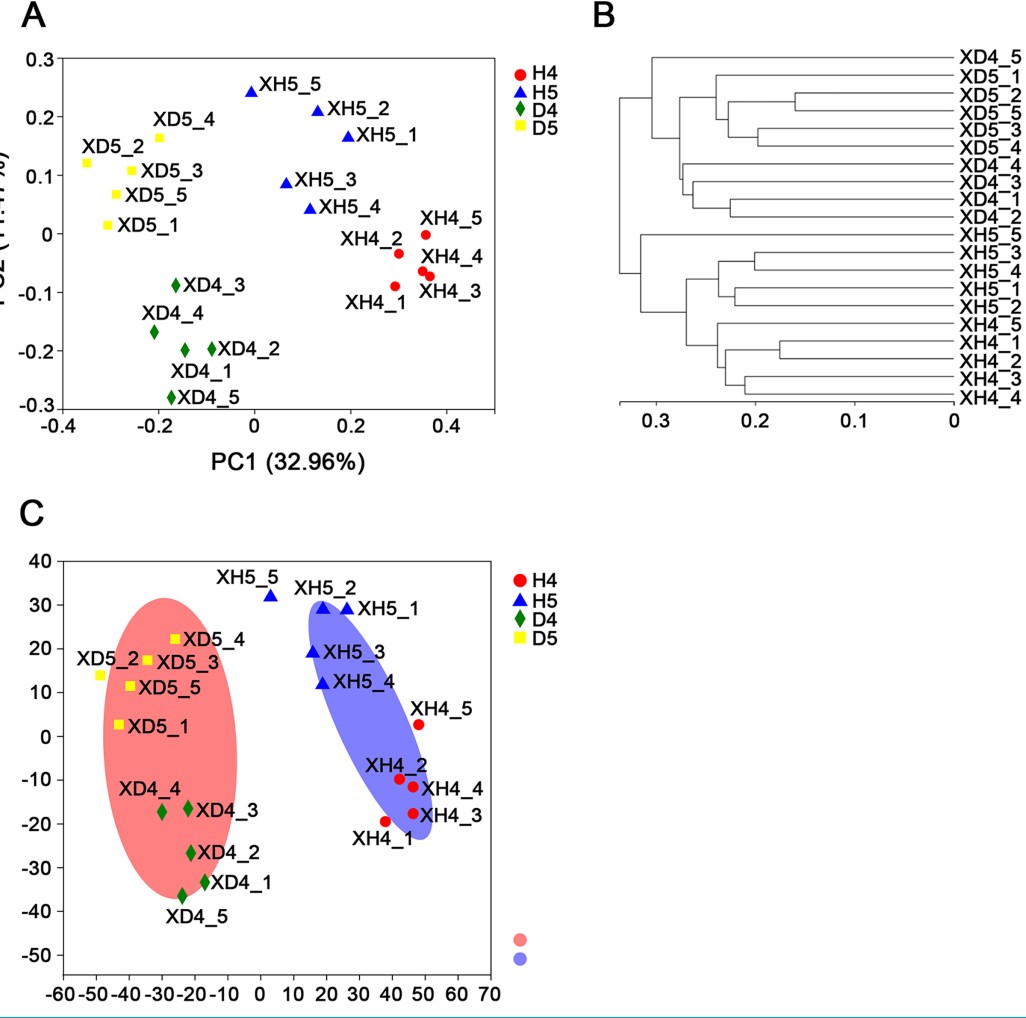

**Figure 2 Comparative analysis of OTUs and genus levels in each community.** (A) Principal coordinate analysis (PCoA) at the OTU level in healthy and diseased groups at the heading stage and filling stage. Distance algorithm is based on bray_curtis. (B) Sample hierarchical clustering analysis based on OTU level. (C) Typing analysis of fungi at the genus level.

and *Blastocladiomycota* (Fig. 2A). Among these phyla, *Ascomycota* was the most abundant. The richness values were 65.48% and 67.61% in healthy groups 4 and 5 (H4 and H5, respectively), and 72.57% and 57.35% in diseased groups 4 and 5 (D4 and D5, respectively), respectively, but there was no significant difference among the populations. The other three phyla showed very interesting results. *Basidiomycota* had the highest abundance in population H4 in the heading stage (25.3%). However, in the D4, D5 and H5 populations, the abundance was significantly decreased to 12.36%, 11.16% and 14.7%, respectively. Significant differences were found between populations H4 and D4 and populations D5 and H5 ($P < 0.01$). The abundance of *Zygomycota* in populations H4 and D4 was low in the heading stage (3.16% and 2.44%, respectively), but suddenly increased to 9.87% (H5) and 7.42% (D5) in the filling stage. However, the trends in the abundance of *Chytridiomycota* and *Zygomycota* were almost the opposite. The abundance of *Chytridiomycota* in the heading stage groups was 4.52% (H4) and 7.18% (D4), and there

**Table 2 All fungi with a richness greater than one percent in each group.** There are 30 genera in the four groups, which mainly belong to *Ascomycota*, *Basidiomycota*, and *Chytridiomycota*.

| Phylum | Genus | H4 | D4 | H5 | D5 |
|---|---|---|---|---|---|
| p__Ascomycota | g__Microdochium | 0.0184 | 0.0760 | 0.0455 | 0.0454 |
| | g__Apodus | 0.0010 | 0.0662 | 0.0035 | 0.0314 |
| | g__Mycosphaerella | 0.1291 | 0.0460 | 0.0609 | 0.0136 |
| | g__Scytalidium | 0.0005 | 0.0309 | 0.0027 | 0.0168 |
| | g__Chaetomium | 0.0028 | 0.0295 | 0.0029 | 0.0132 |
| | g__Alternaria | 0.1489 | 0.0284 | 0.1160 | 0.0288 |
| | g__Aspergillus | 0.0119 | 0.0274 | 0.0021 | 0.0035 |
| | g__Articulospora | 0.0012 | 0.0271 | 0.0044 | 0.0045 |
| | g__Cistella | 0.0010 | 0.0236 | 0.0015 | 0.0164 |
| | g__Acremonium | 0.0103 | 0.0213 | 0.0165 | 0.0081 |
| | g__Epicoccum | 0.0442 | 0.0160 | 0.0201 | 0.0057 |
| | g__Subplenodomus | 0.0005 | 0.0134 | 0.0009 | 0.0001 |
| | g__Debaryomyces | 0.0008 | 0.0108 | 0.0001 | 0.0003 |
| | g__Sarocladium | 0.0193 | 0.0055 | 0.0127 | 0.0042 |
| | g__Fusarium | 0.0021 | 0.0050 | 0.0197 | 0.0100 |
| | g__Gibberella | 0.0122 | 0.0036 | 0.0083 | 0.0038 |
| | g__Cladosporium | 0.0135 | 0.0027 | 0.0103 | 0.0013 |
| | g__Neosetophoma | 0.0106 | 0.0025 | 0.0195 | 0.0008 |
| | g__Monographella | 0.0114 | 0.0003 | 0.0259 | 0.0015 |
| | g__Ilyonectria | 0.0025 | 0.0001 | 0.0210 | 0.0005 |
| | g__Cladorrhinum | 0.0000 | 0.0000 | 0.0118 | 0.0000 |
| p__Basidiomycota | g__Agrocybe | 0.0000 | 0.0255 | 0 | 0.0006 |
| | g__Cryptococcus | 0.0726 | 0.0102 | 0 | 0.0161 |
| | g__Ceratobasidium | 0.0009 | 0.0022 | 0 | 0.0109 |
| | g__Psilocybe | 0.0049 | 0.0013 | 0 | 0.0214 |
| | g__Volvopluteus | 0.0156 | 0.0011 | 0 | 0.0001 |
| | g__Cystofilobasidium | 0.0319 | 0.0009 | 0 | 0.0003 |
| | g__Coprinopsis | 0.0113 | 0.0006 | 0 | 0.0001 |
| p__Chytridiomycota | g__Olpidiaster | 0.0432 | 0.0529 | 0 | 0.0128 |
| | g__Mortierella | 0.0315 | 0.0244 | 0 | 0.0737 |

was a significant difference between these two groups. The abundance of *Chytridiomycota* in population D4, a diseased group, was much higher than that in the healthy groups. By May, the abundance of *Chytridiomycota* in population H5 decreased sharply to 1.37%, while that of population D5 decreased to only 4.12%.

Excluding unclassified fungi, all fungi with richness greater than 1% in a single group were counted (Table 2). There are 30 genera in the four groups belonging to *Ascomycota*, *Basidiomycota*, *Chytridiomycota* and *Zygomycota*. Among these genera, *Chytridiomycota* and *Zygomycota* had only one genus, *Olpidiaster* and *Portierella*, respectively. *Basidiomycota* had seven genera, and *Ascomycota* had the largest distribution, with 21 genera (Figs. 2B and 2C; Table 2). In the heading stage, the abundance of the most

highly abundant genera in population D4 was significantly decreased in population D5 at the filling stage; however, the abundance of *Mortierella* significantly increased, the abundance of one-third of the genera remained almost unchanged, and the abundance of a few low abundance genera, such as *Fusarium* and *Ceratobasidium*, significantly increased in the diseased population over time. Seven genera with high richness were found in the H4 population at the heading stage, and the richness of these genera was significantly increased in the H5 population at the grain filling stage; among these genera, the variation in the richness of *Microdochium*, *Mycosphaerella*, *Scytalidium*, *Acremonium* and *Olpidiaster* from the H4 population to the H5 population was the opposite of the change in abundance from the D4 population to the D5 population. The richness of 13 genera did not markedly change, and the richness of four low-richness genera was significantly increased at the grain filling stage compared with the heading stage. In general, the following six genera showed no change in richness from the heading stage to the grain filling stage, between the H5 and D5 populations and the H4 and D4 populations, respectively, *Alternaria*, *Sarocladium*, *Gibberella*, *Cladosporium*, *Neosetophoma* and *Cystofilobasidium*. The genera with significant or extremely significant differences in richness among the groups were *Alternaria*, *Mortierella*, *Cryptococcus*, *Apodus*, *Epicoccum*, *Scytalidium* and *Chaetomium*.

A Venn diagram of genera with richness greater than 1% was constructed (Fig. 2D). There were 153 shared genera, 35 endemic genera that appeared most frequently in the healthy group in the filling stage and seven and 10 genera in the heading stage and filling stage, respectively. The results showed that the root exudates of diseased plants specifically promoted the growth of microbial fungi, while the rhizosphere of healthy plants was more suitable for the growth of various fungi. An increased diversity of soil microorganisms promoted healthy plant growth.

## PCoA of populations

The PCoA of fungal community structure in different samples based on OTU revealed that the populations could be divided into two groups: the healthy group and the diseased group (Figs. 3A and 3B). There was no overlap between the two groups, which indicated that the samples were consistent with the expectation and that the microbial community structures of the diseased group and the healthy group were quite different. The diseased group at the heading stage was significantly separated from the diseased group at the filling stage, and the microbial community structure was quite different. The healthy group at the heading stage and the healthy group at the filling stage were also clearly separated, and the microbial community structure was quite different. In addition, according to the flora classification, all rhizosphere fungi were divided into two types (Fig. 3C): type one included the diseased group, and type two included the healthy group. Only sample H5-5 was assigned to type one.

## Effects of physical and chemical soil properties on rhizosphere fungi

The main physical and chemical properties of soil measured were total phosphorus (TP), ammonium nitrogen ($NH_4$), nitrate nitrogen ($NO_3$), total nitrogen (TN), pH, soil density
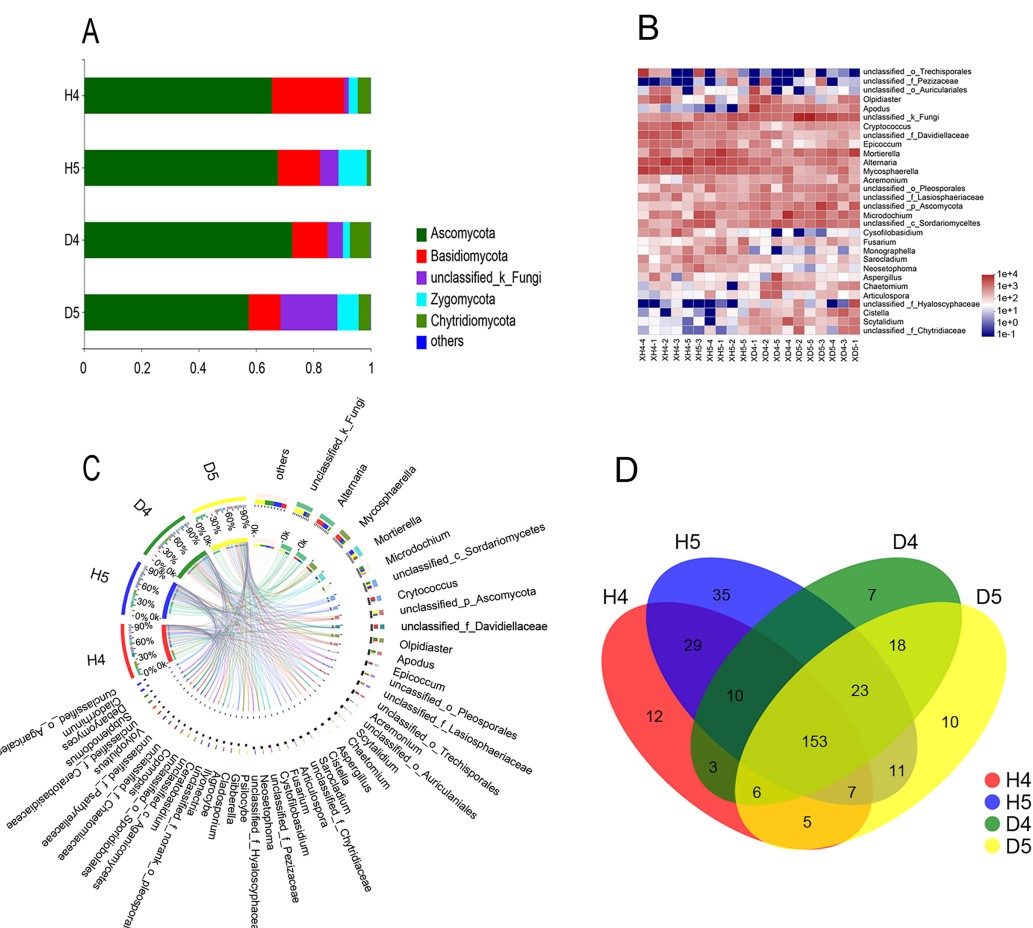

**Figure 3 Species composition analysis.** (A) Percent community abundance at the phylum level in each groups. (B) Species abundance clustering at genus level in each sample. (C) Circos representation showing distribution of genus with significant difference in abundance in different groups. (D) Analysis of common and endemic genera in the different populations by Venn diagram.

**Table 3 The physical and chemical properties of the rhizosphere.**

| Sample ID | TP | NH$_4$ | NO$_3$ | TN | pH | SD | TC | DHR |
|---|---|---|---|---|---|---|---|---|
| H4 | 4.16 ± 0.13 | 16.02 ± 3.27 | 26.38 ± 6.03 | 204.1 ± 30.94 | 7.18 ± 0.06 | 2.68 ± 0.06 | 9.82 ± 1.37 | 86.26 ± 1.32 |
| D4 | 4.13 ± 0.21 | 14.04 ± 2.62 | 30.97 ± 5.95 | 251.48 ± 21.09 | 6.95 ± 0.08 | 2.42 ± 0.05 | 10.56 ± 2.25 | 85.15 ± 0.84 |
| H5 | 3.82 ± 0.18 | 17.45 ± 4.20 | 23.48 ± 2.23 | 217.65 ± 49.43 | 7.15 ± 0.07 | 2.5 ± 0.03 | 7.93 ± 0.71 | 89.77 ± 0.32 |
| D5 | 3.84 ± 0.16 | 18.71 ± 2.41 | 29.38 ± 3.82 | 237.39 ± 52.74 | 6.76 ± 0.15 | 2.48 ± 0.03 | 8.54 ± 0.43 | 88.77 ± 0.21 |

**Note:**
TP, total phosphorus; NH$_4$, ammonium nitrogen; NO$_3$, nitrate nitrogen; TN, total nitrogen; SD, soil density; TC, total carbon; DHR, soil dry-humidity ratio.

(SD), total carbon (TC) and the soil dry-humidity ratio (DHR) (Table 3). TP and TC decreased and NH$_4$ increased from the heading stage to the filling stage. Moreover, the NO$_3$ and TN of the diseased group were higher than those of the healthy group, and the pH value and SD of the diseased group were lower than those of the healthy group. Heatmap cluster analysis based on Spearman correlation coefficients for the 30 genera with

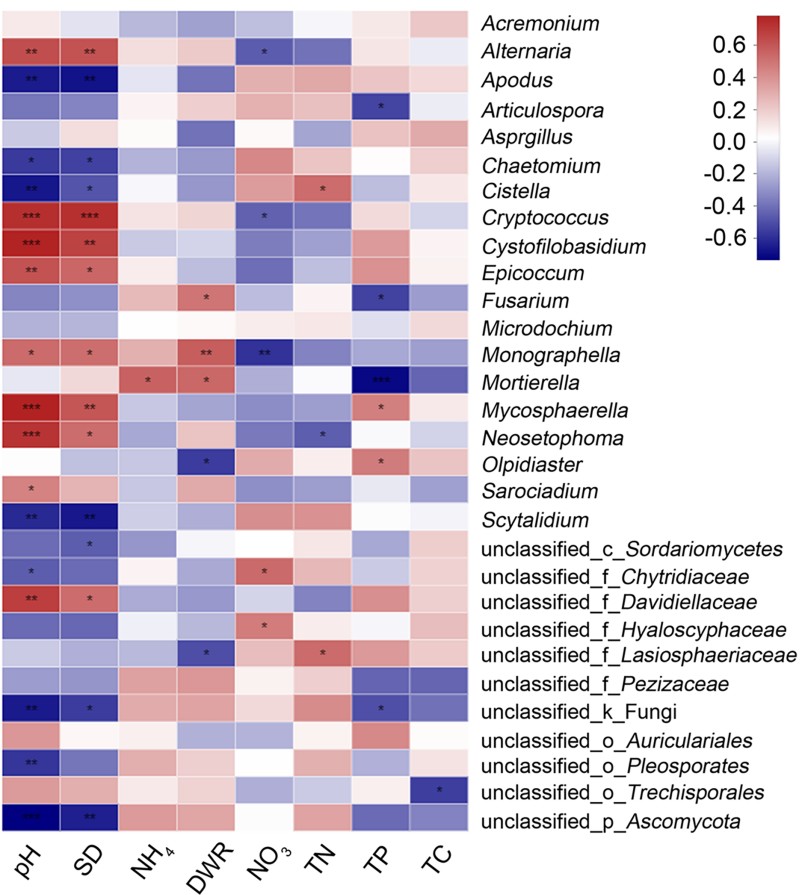

**Figure 4 Hierarchical clustering analysis at the OTU level between physical and chemical soil properties and the 30 most abundant genera.** *, ** and *** represent significant difference *P* < 0.05, *P* < 0.01 and *P* < 0.001, respectively.

the most richness and these physical and chemical indicators was performed (Fig. 4). The graph shows that pH and SD have the greatest influence on microbial abundance, and the influence of these factors is in the same direction. Soil density refers to the dry weight of unit volume soil (g/cm$^3$) (not including soil pore), which also represents water saturation. Soil density can affect the soil microbial diversity and community structure. One-third of the genera were positively or significantly positively correlated with both pH and SD, and one-third were negatively or significantly negatively correlated with both pH and SD. The effects of NO$_3$ and TN on microbial richness were similar, but the impact of these factors on microbial species richness was almost the opposite of that of pH and SD. From our results, there is relationship between them, but from previous similar research, there is no necessary reverse relationship between them. Soil DHR and TP had moderate effects on microbial richness. The physical and chemical properties of soil with the least influence on microbial abundance were NH$_4$ and TC.

## DISCUSSION

Microbial diversity in soil is an important factor that determines soil health and is considered one of the main contributors to soil suppressiveness (*Xu et al., 2012*; *Miao et al., 2016*). The rhizosphere is one of the most complex environments; rhizospheres are influenced by plant roots and are an active microhabitat where plant roots and microbes interact (*Xu et al., 2012*; *Mendes, Garbeva & Raaijmakers, 2013*; *Singh, Singh & Dubey, 2014*). Throughout the heading and filling stages of wheat, the fungi in the rhizosphere soil showed high diversity. The diversity of soil fungi at the grain filling stage was significantly higher than that at the heading stage. It is already well known that most crops can significantly benefit from establishing associations with diverse soil microbes (*Kristin & Miranda, 2013*). Plants stimulate or inhibit the growth of specific rhizosphere microorganisms by releasing secondary metabolites into the rhizosphere (*Chakraborty et al., 2011*; *Meena, Rakshit & Meena, 2016*; *Kumar et al., 2017*). For example, Flavonoids can resist root bacteria (there are many studies on fungi) and play a bactericidal effect in the way of cell death: compounds are oxidized to release toxic active ions leading to cell death; Degradation of ATPASE and mitochondrial respiratory electron transporter on the membrane to achieve cell death (*Zhalnina et al., 2018*). In the rhizosphere soil of wheat root rot disease, the interaction between wheat and microorganisms was intense, and the diversity and richness of the fungal community in the rhizosphere soil of the diseased group were significantly higher than that of the healthy group, creating conditions for the occurrence of wheat root rot. However, at the filling stage, wheat roots were clearly diseased due to infection by pathogenic fungi. During this process, the metabolites secreted by wheat roots tend to be less (*Chen, Waghmode & Sun, 2019*). As a result, the fungi in the rhizosphere of diseased plants showed lower community diversity and richness than those in the rhizosphere of healthy plants. The interaction between rhizosphere fungi and wheat roots in the healthy group reached an optimal balance during the filling stage and then became mutualistic, which was favorable for the healthy growth of the fungi and wheat. Therefore, in the filling stage, the diversity and richness of fungi in the rhizosphere were higher in the healthy group than in the diseased group.

The richness change analysis at the phylum level revealed that the phylum with the highest richness was *Ascomycota*, which consisted of more than 60% of all healthy and diseased groups, and there was no significant difference among groups. However, the main pathogenic fungi that cause wheat root rot disease, such as *F. culmorum*, *F. pseudograminearum*, *G. graminis* var. *Tritici*, *B. sorokiniana* and *Alternaria* spp., are in this phylum. There are also some fungi that cause wheat root rot in the phylum *Basidiomycota*, such as *Rhizoctonia oryzae*, *R. solani* and *Penicillium* spp (*Almasudy, You & Barbetti, 2015*; *Barnett, Ballard & Franco, 2019*; *Gqozo et al., 2020*; *Zhang, Yu & Wang, 2021*). This phylum showed high richness in the healthy group at the heading stage (25.3%), but the richness of this phylum was significantly decreased in the other groups (<14%). Analysis of the abundance of these two phyla shows that the impact of the fungi that cause wheat root rot disease is not due to phylum-level abundance. The richness of two other phyla, *Zygomycota* and *Chytridiomycota*, which contain almost no root rot
fungi, showed notable differences between the heading and filling stages in the healthy group but showed no significant differences between these stages in the diseased groups. This change in abundance may contribute to the healthy growth of wheat roots or may be a cofactor in the occurrence of wheat root rot disease.

Relative to the changes at the phylum level, levels of richness at the genera level varied greatly. The abundance of most genera with high richness decreased significantly from the heading stage to the filling stage in the diseased groups; the richness of approximately one-third of all genera remained unchanged, and only very few low-richness genera, such as *Fusarium* and *Ceratobasidium*, had a very significant increase in richness over time. In the healthy group, the abundance of most genera increased significantly from the heading stage to the filling stage, except for some genera whose abundance did not markedly change or very few genera whose abundance increased significantly. This result also shows that the interactions between wheat roots and rhizosphere fungi must achieve a balance. If this balance is lost, the wheat roots will become diseased. From the analysis of the endemic genera in each group, we also found that there were 35 endemic genera in the healthy group, which was many more than the approximately 10 endemic genera in the diseased group, indicating that the healthy growth of wheat roots can promote plant growth and suppress disease through various activities that prevent infection by pathogens. Therefore, the rhizosphere microorganisms show improved growth, and the microbial diversity and community richness are also significantly increased in the healthy plants compared with the diseased plants.

Fungi and fungus-like organisms form one of the most diverse groups of eukaryotes and represent an essential functional component of soil microbial communities (*Buée et al., 2009*; *Miao et al., 2016*). Under unfavorable conditions, some fungi can cause plant diseases and sometimes even the total loss of crop yields. In many instances, these diseases are caused by a complex of fungal species (*Miao et al., 2016*). Only *Alternaria* and *Fusarium*, genera with richness values greater than one percent, have been documented to cause wheat root rot, but the abundance of these two genera is not high. Although the abundance of *Alternaria* in the diseased groups was significantly higher than that in the healthy groups, there was no significant change between the heading stage and filling stage. The abundance of *Fusarium* in the filling stage was significantly higher than that in the heading stage. Considering the abundance of these two pathogens, the richness level of a fungus does not indicate whether it can lead to wheat root rot. In addition, other pathogenic fungal genera mentioned in the literature were determined to have less than one percent abundance. Therefore, we can speculate that as long as these fungal genera exist in the rhizosphere, they can lead to wheat root rot disease, regardless of the abundance. However, whether a fungus can induce root rot disease depends on the result of the interaction between wheat root and rhizosphere microorganisms, which is also closely related to seasonal climatic conditions (*Campanella et al., 2020*).

Differences in the fungal community structure among groups can clearly demonstrate the heterogeneity of each group, thus showing why there are so many differences in the rhizosphere fungi of the diseased groups and the healthy groups (*Karuppiah et al., 2020*). PCoA clustering analysis clearly clustered the four groups into two groups: the

healthy group and the diseased group. However, the healthy group and the diseased group were each clearly separated into the heading stage and grouting stage. In addition, all rhizosphere fungi were divided into two types according to flora classification: one type included only the healthy group, and the other type included only the diseased group. When wheat roots were attacked by microorganisms and developed root disease, the structure of the rhizosphere fungal community was markedly changed (*Stephen, Ross & Christopher, 2019*). To differentiate between the fungi located very close to the epidermis in the root zone and to protect against the invasion of heterogeneous microbes, plants continuously secrete signaling molecules, which allows for the development of pathogenic, associative, symbiotic, or naturalistic relationships between microbes and the plant (*Kumar et al., 2017*; *Hayat et al., 2010*).

At the heading stage, the root is slightly diseased, and the interaction between wheat roots and rhizosphere fungi is the most intense, resulting in marked heterogeneity of the soil environment and inducing a high level of rhizosphere fungal diversity and a complex community structure. However, at the filling stage, the wheat roots are completely diseased. At this time, the soil environment is stable, and the material secreted by wheat roots is relatively limited, which leads to a decrease in the microbial diversity of the rhizosphere and a relatively simple community structure. Soil-plant-microbial health must be maintained at an equilibrium to maintain sustainable agricultural practices (*Narula, Anand & Dudeja, 2013*; *Ramırez-Bahena et al., 2013*; *Kumar et al., 2017*). At the heading stage, the antagonistic interaction between healthy wheat roots and rhizosphere fungi reaches a balance and promotes the healthy growth of wheat and fungi. By the filling stage, the ecological environment, such as temperature, humidity and other factors, is improved, the rhizosphere microbial diversity naturally significantly increases, and the community structure becomes more complex. When microbial diversity increases, the beneficial microorganisms will also increase, which will improve the inhibitory effect of plant roots, thus reducing the occurrence of plant root rot (*Hu et al., 2016*).

Biotic and abiotic factors are assumed to influence the structural and functional diversity of the microbial communities in the rhizosphere (*Berg & Smalla, 2009*; *Weinert et al., 2011*). Site properties, including soil type, climatic conditions and type of agricultural management, have been shown to strongly influence the relative composition of rhizosphere microbial communities (*Heuer et al., 2002*; *Kowalchuk et al., 2002*; *Berg et al., 2006*; *Costa et al., 2006*; *Bremer et al., 2007*; *Weinert et al., 2010*). Changes in the physical and chemical properties of soil also have a significant impact on fungal diversity and community structure in crop rhizospheres. Soil physical and chemical properties are determined not only by the nature of the soil itself but also by the physical and chemical properties of the soil after the interaction between crop roots and rhizosphere fungi. The physical and chemical soil properties we measured were determined in soil collected around the rhizosphere of wheat (*Zhang, Vivanco & Shen, 2017*). Organisms that are present in the rhizosphere microbiota can have profound effects on the growth, nutrition and health of plants in agroecosystems (*Bonfante & Anca, 2009*; *Mendes et al., 2011*; *Meena et al., 2015*; *Kumar et al., 2017*). TP and TC decreased from the heading stage to the filling stage, indicating that the later the growth period was, the less TP and TC was

needed; however, these metrics seemed unrelated to the occurrence of diseases.

In addition, the levels of $NH_4$, $NO_3$ and TN in the diseased group were higher than those in the healthy group at the heading and filling stages, indicating that their increase contributed to the occurrence of wheat root rot disease. Too much ammonium nitrogen and total nitrogen can easily damage the root system of the plant, then it can easily attract pathogenic fungi to attack the damaged root system of the plant, resulting in the occurrence of root to disease (*Zhang et al., 2021*). In contrast, the pH and SD of the soils around diseased plants were lower than those of the soils around healthy plants at the heading and filling stages. This finding indicated that relatively low pH and SD values were beneficial to the occurrence of wheat root rot disease. The heatmap cluster analysis of the physical and chemical soil indicators and the 30 most abundant genera also showed that soil pH and SD affected fungal abundance and diversity in the same direction.

## CONCLUSIONS

In the rhizosphere, many plant-microbial interactions occur that mediate soil processes (*Kumar et al., 2017*). The occurrence of crop root rot disease is closely related to the interaction between rhizosphere microorganisms and crop roots, as well as the physical and chemical properties of soil. At present, there are approximately 10 types of fungi that can cause wheat root rot disease alone or in a complex, according to the literature.

By studying the diversity of fungi and the community structure of the rhizospheres of healthy and diseased wheat at different growth stages, the heading and filling stages, it was revealed that in the early stages of illness, the high diversity of rhizosphere fungi and a complex community structure can easily cause wheat root rot disease. Additionally, the existence of pathogenic fungi is a necessary condition for wheat root rot disease, but the richness of pathogenic fungi is not necessarily important. Based on the physical and chemical properties of the soil, an increase in $NH_4$, $NO_3$ and TN contributes to the occurrence of wheat root rot disease. Soil pH and SD had the greatest influence on the abundance and diversity of rhizosphere fungi, and the influence was in the same direction; low soil pH and SD are beneficial to the occurrence of wheat root rot disease.

### Funding

This work was financially supported by the National Key Research and Development Program of China (2017YFD0200605), the Key Research and Development Project (2017YFD0201600), and the Key Technology Research and Demonstration Project of Hubei Agricultural Science and Technology Innovation Center (2020-620-000-002-07). The funders had no role in study design, data collection and analysis, decision to publish, or preparation of the manuscript.

### Grant Disclosures

The following grant information was disclosed by the authors:
National Key Research and Development Program of China: 2017YFD0200605.

Key Research and Development Project: 2017YFD0201600.
Key Technology Research and Demonstration Project: 2020-620-000-002-07.

## Competing Interests

The authors declare that they have no competing interests.

## Author Contributions

- Xuejiang Zhang conceived and designed the experiments, performed the experiments, analyzed the data, prepared figures and/or tables, authored or reviewed drafts of the paper, and approved the final draft.
- Heyun Wang performed the experiments, prepared figures and/or tables, authored or reviewed drafts of the paper, and approved the final draft.
- Yawei Que conceived and designed the experiments, performed the experiments, analyzed the data, prepared figures and/or tables, authored or reviewed drafts of the paper, and approved the final draft.
- Dazhao Yu conceived and designed the experiments, analyzed the data, authored or reviewed drafts of the paper, and approved the final draft.
- Hua Wang conceived and designed the experiments, analyzed the data, authored or reviewed drafts of the paper, and approved the final draft.

## Field Study Permissions

The following information was supplied relating to field study approvals (*i.e.*, approving body and any reference numbers):

The Institute of Plant Protection and Soil & Fertilizer, Hubei Academy of Agricultural Sciences (Project number:17.035.18).

## Data Availability

The sequences are available at NCBI Bioproject: PRJNA549031.

## Supplemental Information

Supplemental information for this article can be found online at http://dx.doi.org/10.7717/peerj.12601#supplemental-information.

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
