# Peer review of "The influence of rhizosphere soil fungal diversity and complex community structure on wheat root rot disease"

_PeerJ, doi:10.7717/peerj.12601_

## Round 0.1 · original submission · Minor Revisions

Two experts revised your manuscript and found your data interesting and with the potential to be published in this journal. Please address all the concerns included in this message.

·

Basic reporting

General comments

This is an interesting manuscript in which the authors report a correlation between soil fungal community richness and symptomatic or non-symptomatic plants at the heading and filling stages of hexaploid wheat. However, there are inconsistencies between the importance of soil fungal diversity as mentioned in the Introduction and the analyses reported in Results and Discussion. It would be interesting to see the diversity if the OTUs were parsed by year as well as by host developmental stage. The Discussion is redundant with Results in some places (e.g., L352-359). The authors should consider discussing what the results mean, and what might be the next steps to explore. Also, the manuscript lacks certain details that would improve the utility and significance of the data to the broad readership of PeerJ, as described in Basic Reporting and Experimental Design.

Basic Reporting

The authors mention the importance of soil microbial diversity and soil structure to sustainable agriculture in the Introduction but their annotation to the genus level does not distinguish between beneficial and wheat-pathogenic fungal species within the genus. In this reviewer’s opinion, soil health can indirectly be conditioned by crop and crop rotation, root exudate quality and environmental factors.

Is there any information on the types of root rot observed at the Xiangyang Original Farm in previous years. For instance, what information is available for the causal agents in the diseased plots.

The authors should state whether alpha diversity was analyzed using R, and describe how the soil physical and chemical properties were quantified.

References for data processing steps will be useful to readers who have not used Trimmomatic or FLASH.

Do the statements in the Discussion L292-297 refer to references cited immediately before? If not, please provide new references.

The English grammar and spelling are excellent. However, certain words are unclear, such as exhaustive (L57); ears on L 136 (do you mean heads); endemic (L248); monotonous (L297).

L72 – Include Rhizoctonia oryzae or omit R. solani and substitute Rhizoctonia spp.
L90 – Nutrients are also drivers for rhizosphere soil community structure.
L199-200 - High community richness can also be a factor in disease suppression.
L273 – Explain briefly what soil density represents—water saturation, or clay/sand composition?
Fig. 1 – Describe to the readers what the plants are showing. Any stunting, chlorosis?
Figures 2 and 3 appear to be switched as presented.

Experimental design

The author should clarify how the 100 samples were selected for the 5 sequencing reps. Were any samples pooled before extraction or the extracts pooled for sequencing.

The authors could consider identify the OTUs to species using UNITE and Blastn, especially to look for known soilborne pathogen species of the cereals.

Validity of the findings

The findings have moderate validity regarding the correlation between host floral/head development and soil community members.

The raw data deposition to is valid. Raw data on the soil properties is provided in a supplementary table.

Reviewer 2 ·

Basic reporting

In general, the research has been carried out with scientific rigor, the proposed methods are adequate to achieve the goals.

Experimental design

The methods are adequate, as well as the organization of the manuscript.
There are some suggestions, I will show them shortly:

Line 72: ;Rhizoctonia is spp or what specie?
Line 100: microflora --> microbiota
Line 125: Reference the method of collecting the samples or mention if it is from this study.
Line 136-137: Although it is not totally necessary, you could place an image to make the comparison between the diseased and healthy plant (perhaps in supplementary material).

Fig. 2. If possible, enlarge panels B and C; since it is difficult to visualize them.

Lines 216-224: How would the differences that exist in that short time be explained?

What explanation exists for this relationship that exists between a greater quantity of total nitrogen and a greater quantity of pathogenic fungi?

Line 277: Why is there that relationship?
Line 291: What are some examples of these metabolites?
Line 297: Is there any reference for this statement?
Line 309-310: Reference
Line 377-380: So if microbial diversity increases, would it tend to a balance between species and therefore a decrease in infection?

Line 387-390: Reference.

Line 397-398: Is there an explanation for this? Or is there any reference to it?

Validity of the findings

The proposed conclusions are supported by the results achieved.

Some references are needed, I indicated them previously.

Additional comments

Review the suggestions shown in the previous sections.

---

## Round 0.2 · accepted · Accept

The manuscript was significantly improved, following the Reviewers' comments.